# Cytokine Release Syndrome Associated with T-Cell-Based Therapies for Hematological Malignancies: Pathophysiology, Clinical Presentation, and Treatment

**DOI:** 10.3390/ijms22147652

**Published:** 2021-07-17

**Authors:** Maria Cosenza, Stefano Sacchi, Samantha Pozzi

**Affiliations:** Department of Medical and Surgical Sciences, University of Modena and Reggio Emilia, 41124 Modena, Italy; stefano.sacchi@unimore.it (S.S.); samantha.pozzi@unimore.it (S.P.)

**Keywords:** cytokine release syndrome, CAR T cell therapy, monoclonal antibodies, hematological malignancies

## Abstract

Cytokines are a broad group of small regulatory proteins with many biological functions involved in regulating the hematopoietic and immune systems. However, in pathological conditions, hyperactivation of the cytokine network constitutes the fundamental event in cytokine release syndrome (CRS). During the last few decades, the development of therapeutic monoclonal antibodies and T-cell therapies has rapidly evolved, and CRS can be a serious adverse event related to these treatments. CRS is a set of toxic adverse events that can be observed during infection or following the administration of antibodies for therapeutic purposes and, more recently, during T-cell-engaging therapies. CRS is triggered by on-target effects induced by binding of chimeric antigen receptor (CAR) T cells or bispecific antibody to its antigen and by subsequent activation of bystander immune and non-immune cells. CRS is associated with high circulating concentrations of several pro-inflammatory cytokines, including interleukins, interferons, tumor necrosis factors, colony-stimulating factors, and transforming growth factors. Recently, considerable developments have been achieved with regard to preventing and controlling CRS, but it remains an unmet clinical need. This review comprehensively summarizes the pathophysiology, clinical presentation, and treatment of CRS caused by T-cell-engaging therapies utilized in the treatment of hematological malignancies.

## 1. Introduction

Cytokines are small molecular messengers produced by a wide variety of immune and non-immune cells [1,2,3,4] that act as mediators and modulators within microenvironments. Cytokines regulate immunological responses, hematopoietic development, and cell to cell communication, as well as host responses to infectious agents, inflammatory stimuli, and drugs, modulating their effects [4,5,6]. Cytokines with important roles in the hematopoietic and immune systems may be classified based on their structure or function as interleukins (ILs), interferons (IFNs), tumor necrosis factors (TNFs), colony-stimulating factors (CSFs), and transforming growth factors (TGFs) [7]. Under physiological conditions, the secretion of cytokines is highly regulated, and the excess production of one cytokine is antagonized by the production of others with opposing functions through a counter-regulatory homeostatic mechanism. Cytokines can be pleiotropic, with different effects on diverse cell types, and can act synergistically. They form complex interactive networks with potential autocrine, paracrine, and endocrine functions [4,8,9]. Cytokine release syndrome (CRS) is a systemic inflammatory response that can be triggered by a variety of factors, such as infection and certain medications, including monoclonal antibodies. CRS has been described after the infusion of several antibody-based therapies, including rituximab [10,11], obinutuzumab [12], alemtuzumab [13], brentuximab [14], dacetuzumab [15], and nivolumab [16]. Severe viral infections, such as influenza and SARS-CoV-2 (COVID-19) [17], can also trigger CRS through massive immune and non-immune cell stimulation.

Several efforts have been made to identify new therapeutic strategies for treating hematological malignancies, and some immunotherapy approaches have been tested to fortify the immune system of the patient against tumors. In the last few years, immune checkpoint inhibition and T-cell-engaging therapies, such as bispecific T-cell-engaging (BiTE) single-chain antibody constructs and chimeric antigen receptor (CAR) T cells, have opened up a new frontier in cancer immunotherapy [18,19,20]. However, one of the most important serious adverse effects of these therapies is CRS.

CRS is characterized by hypersecretion of pro-inflammatory cytokines, including IL-6, IL-1, IL-5, IL-10, IFN-γ, TNF, and TGFs by B and T lymphocytes and natural killer (NK) cells. The CRS may be further enhanced by numerous cellular interactions with bystander cells, such as endothelial cells, monocyte/macrophages, and dendritic cells, further increasing cytokine hypersecretion, aggravating symptoms, and inducing various grades of organ damage [21]. CRS symptoms may occur immediately after the administration of T-cell-engaging therapies or may be delayed until days or weeks after treatment. CRS can manifest as mild, with flu-like symptoms, including fever, nausea, and chills, or may be life-threatening and severe with shock and respiratory compromise, leading to multi-system organ failure and even death [10,22].

This review comprehensively summarizes the biological and clinical aspects of the CRS triggered by T-cell-engaging therapies used in the treatment of hematological malignancies.

## 2. Pathophysiology

The pathophysiology of CRS has been associated with invasive pathogens and therapeutic infusions of several monoclonal antibodies [10,12,13,14,15,23] (Table 1). CRS can also develop in association with severe viral infections, including COVID-19, which is caused by SARS-CoV-2 [24,25]. Recently, with the success of the newer T-cell-engaging immunotherapeutic agents, such as BiTE constructs and CAR T cells, in hematological malignancies [26], the interest in CRS has grown, as this is a major serious adverse event of these treatments. The immunotherapeutic strategies have been carried forward into clinical applications and shown impressive therapeutic activity in several hematological malignancies, including acute lymphoblastic B cell leukemia (B-ALL), chronic lymphocytic leukemia (CLL), and diffuse large B cell lymphoma (DLBCL) [27]. CRS describes an exaggerated systemic immune response involving the release of more than 150 inflammatory mediators, cytokines, chemokines, oxygen radicals, and complement factors (Table 2) [28]. CRS is due to on-target effects induced by the binding of CAR T cells or BiTE antibody to its antigen on the surface of target cells and subsequent activation of bystander immune and non-immune cells, such as monocytes/macrophages, dendritic cells, and endothelial cells. Activation of these cells results in the massive release of several cytokines, initiating a cascade of events that overwhelms counter-regulatory homeostatic mechanisms, leading to CRS [29,30]. As clearly illustrated in Figure 1, T-cell engaging therapies target tumor cells and induce the release of cytokines as IFN-γ or TNF-α, which lead to the activation of bystander immune and non-immune cells as monocytes/macrophages, dendritic cells, NK and T-cell, and endothelial cells. These cells further release proinflammatory cytokines triggering a cascade reaction. Macrophages and endothelial cells produce large amounts of IL-6 which in turn activates T cells and other immune cells leading to a cytokine storm (Figure 1). However, the pathophysiology of CRS is still poorly understood.

### 2.1. The Key Role of IL-6

Physiologically, cytokines play a key role in coordinating effector cells of the immune system and providing regulatory signals that direct, amplify, and resolve the immune response. Cytokines have short half-lives, which normally prevents them from having effects outside the lymphoid tissue and sites of inflammation. In CRS, immune over-activation occurs as a result of perceived danger, resulting in excessive activation of effector immune cells and prolonged immune activation.

The overabundance of cytokines causes clinically significant collateral damage. The cytokines involved in the pathophysiology of CRS include IFN-γ, TNF, IL-6, and IL-10, which are consistently found to be elevated in the serum of patients with CRS [22,48] (Figure 1). IL-6 seems to play a key role in CRS pathophysiology, as highly elevated IL-6 levels are found in almost all patients with CRS [10,35]. IL-6 is a pleiotropic cytokine that exhibits both anti-inflammatory and pro-inflammatory characteristics. IL-6 can play diverse roles in different phases of inflammation, promoting damaging reactions within the tissue, and then contributing to resolving the inflammation, and finally helping to repair tissue in the late stages [49]. IL-6 is secreted by T lymphocytes, monocytes/macrophages, dendritic cells, mesenchymal cells, and osteoblasts and is present in processes, such as neutrophil migration, the acute phase response, angiogenesis, B-cell differentiation, and antibody generation [50] (Figure 2A). IL-6 exerts its biological functions via two major pathways: “classical signaling” and “trans-signaling pathways.” In “classical signaling” pathways, IL-6 activates cells by binding to the IL-6 receptor (IL6R), which leads to dimerization of the membrane protein gp130 [50] and intracellular tyrosine kinases, especially Janus kinase (JAK) 1 and JAK2, which then activate transcription factors signal transducer and activator of transcription (STAT) 1 and STAT3 [51,52]. Other cascades that can be activated by IL-6 include the SHP-2/ERK MAPK, PI3K-AKT-mTORC1, and SRC-YAP-NOTCH pathways [53,54,55] (Figure 2B). IL-6 can bind cells that express IL6R and signal transducer membrane protein gp130 on their surfaces. IL6R is also found in its soluble form in body fluids (sIL6R). When IL6 attaches to sIL6R and starts the signaling cascade by binding to cells that express gp130 alone (without IL6R on their surface), it is known as “trans-signaling” [56]. The mode of signaling influences the result: “classical IL-6 signaling” reduces inflammation, whereas “IL-6 trans-signaling” promotes T-cell migration, decreases apoptosis, enhances cytotoxicity, and suppresses regulatory T-cell differentiation [57,58,59]. “Classical IL-6 signaling” does not seem to affect CAR T cell efficacy [60], but the impact of IL-6 trans-signaling on CAR T cells has not been described.

### 2.2. The Roles of Other Cytokines

IL-6 determines the release of other cytokines, such as IFN-γ and TNF-α, by lymphocytes, monocytes, and neutrophils (Figure 1 and Figure 2A). The release of TNF-α and IFN-γ within 1–2 h is followed by an increase in IL-6 and IL-10, and in some cases, of IL-8 and IL-2. IFN-γ generated by T cells and NK cells, or by the tumor cells themselves, is a pro-inflammatory cytokine that activates other immune cells, such as macrophages [61], which produce excessive quantities of additional cytokines [62]. Moreover, IFN-γ triggers macrophage activation, leading to the secretion of host cytokines, including IL-6, TNF-α, and IL-10 [22], which could further intensify the CRS. IL-10 fails to control this process, though it suppresses cellular immunity. Other cytokines have also been found to be elevated during the course of CRS, including IL-1, IL-2, IL-8, IL-5, monocyte chemoattractant protein 1 (MCP-1), and granulocyte-macrophage colony-stimulating factor (GM-CSF), driving some of its manifestations (Figure 1 and Table 2) [22,48,63,64,65,66,67,68]. IL-1 has one of the simplest signaling mechanisms in the innate immune system. It is capable of sensing an infection and triggering an inflammatory response [69]. IL-1 is released from activated macrophages and monocytes, further stimulating the release of IL-6 and inducing nitric oxide synthetase [48]. Endothelial cell activation also plays a role in CRS. Ang-2, a typical marker of endothelial cell activation, is a hallmark of severe CRS, showing that the endothelium plays a key role in the pathophysiology of CRS by amplifying the inflammatory response [67].

### 2.3. Biomarkers

Various studies indicate a correlation between the severity of the CRS and tumor burden, C-reactive protein (CRP) and ferritin levels, and cytokine levels [29,63,70,71] (Table 2). However, whether the cytokines can act as prognostic factors, especially in patients with CRS, in whom cytokine concentrations may already be high, and in patients with malignancies, is not clear. Comparing the increased cytokine level to the baseline value while simultaneously considering some specific markers and following a series of measurements rather than the amount of a single cytokine has been suggested [72]. Furthermore, the serum cytokine measurement is usually not quickly available in most hospitals. In response to IL-6, the liver produces CRP, and this concentration is easily measured by a rapid and inexpensive assay available in the majority of hospitals. Thus, CRP measurements are widely utilized as a surrogate marker of IL-6 bioactivity [73,74]. However, CRP is not specific to CRS and cannot be used to distinguish inflammation due to infectious or non-infectious disease [75]. An excessive increase in ferritin has also been reported in several patients after CAR T-cell infusion, supporting a relationship between CRS and macrophage activation syndrome (MAS)/hemophagocytic lymphohistiocytosis (HLH). However, in the same experience, ferritin did not show utility in predicting the severity of CRS [29].

## 3. Clinical Manifestations

The incidence of CRS varies with the type of immunotherapy, and it is more frequently observed during CAR T-cell therapy than with bispecific antibody blinatumomab infusion. During T-cell therapies, CRS occurs early during the course of treatment [76]. CRS related to blinatumomab therapy usually occurs during the first cycle of therapy, and typically upon starting the infusion. CRS manifests with a wide variety of signs and symptoms of varying severity. The most important and frequent symptoms are summarized in Figure 3. Fever is usually present, and several other symptoms can mimic infection. The patient’s body temperature quite commonly exceeds 40.0 °C. For this reason, the possibility of infection must be ruled out after appropriate cultures, particularly if the patient is neutropenic. Other constitutional symptoms, such as myalgia and arthralgia, may be present. Any organ can be affected during CRS, and the patient can develop nausea, vomiting, skin rash, hemodynamic instability, and capillary leak syndrome with hypotension and tachycardia, disseminated intravascular coagulation, and neurological toxicity [67]. Neurological toxicity may occur together with other symptoms of CRS or when the other symptoms are disappearing. Neurological toxicity includes headache, confusion, delirium, aphasia, tremor, and seizures. Patients with severe neurotoxicity show signs of endothelial activation, including disseminated intravascular coagulation and increased blood-brain barrier (BBB) permeability, which may allow the entry of high concentrations of systemic cytokines, particularly IFN-γ, inducing brain vascular pericyte stress and the consequent secretion of endothelium-activating cytokines. Adverse neurological events can be fatal in rare cases [77]. Although usually reversible, severe cardiac dysfunction is of particular concern. The pathophysiology of acute cardiac toxicity is not clear but it appears similar to that cardiomyopathy is associated with sepsis [78].

Disseminated intravascular coagulation can occur in patients with CRS, especially in those who develop a CRS of grade ≥4. Consumptive coagulopathy manifests with thrombocytopenia, the elevation of D-dimer, prolongation of prothrombin time (PT) and partial thromboplastin time (PTT), hyperfibrinogenemia, and elevation of endothelial activation markers [67]. CRS symptoms may also mimic MAS/HLH [79,80], and the pathophysiology of the syndromes may overlap [81].

## 4. Management of CRS

The optimal clinical management of CRS is still not well-defined since T-cell-engaging therapies were recently introduced into clinical practice; thus, several questions are still unanswered. A reduced incidence of severe CRS has been obtained with cytoreduction, dose adjustment, and premedication with corticosteroids [82].

In patients receiving T-cell-engaging therapies, the approaches to prevention and treatment of CRS may differ substantially. Blinatumomab has a short half-life (~2 h), and CRS symptoms may resolve quickly by interrupting therapy and starting supportive care with or without additional interventions. However, BiTE constructs can be given repeatedly, whereas CAR T cells are usually administered once, but the side effects after CAR T infusion are difficult to reverse because the infused cells can persist for prolonged periods. Therefore, the management of CRS can be diverse between the two therapeutic modalities. Current treatments are based on the severity of the side effects [29,83] using the grading scheme developed by Lee et al. [29] (Table 3 and Figure 3).

Efficient management of patients requires very strict collaboration between several specialties, such as hematology, neurology, and radiology. Sometimes intensive care unit (ICU) referral should be considered so that mechanical ventilation can be offered when necessary. Patients with grade 1 and 2 toxicity experiencing fever, constitutional symptoms, and moderate hypotension are treated symptomatically with antipyretics and antibiotics for fever, with fluids and low dose vasopressors for hypotension and oxygen supplementation when saturation drops.

After CAR T-cell therapy, fever usually precedes CRS. Therefore, patients who develop persistent fever should be frequently evaluated for signs and symptoms of CRS. If an infection cannot be ruled out, empiric antibiotic therapy should be started.

For severe cases of CRS, the patient should be admitted to the ICU for close monitoring. In addition to aggressive supportive care, steroids and repeated doses of Il-6 inhibitor need to be administered when no improvement is observed. In some resistant cases, other anti-inflammatory agents need to be administered, but clinical experience in this setting is undeveloped [29,80,84,85].

## 5. Treatment of CRS

The therapy for CRS is not yet well defined but is based on the use of steroids and inhibitors of IL-6 activity, as well as IL-1, IFN-γ, TNF-α, and IL-2 inhibitors for unresponsive patients.

### 5.1. Steroids

Clinical experience shows that steroids are an effective treatment for suppressing the excessive inflammatory response and CRS [86]. Opinions differ on the timing and dosing of corticosteroids. Some choose to use corticosteroids as a first-line agent, whereas others do not [29]. However, corticosteroids have generalized effects on the immune system and may also inhibit the anti-tumor efficacy and affect the amplification and persistence of CAR T cells in vivo [71]. Thus, steroids should generally be avoided as first-line treatment but used in resistant patients with severe CRS and given at high doses when it is necessary to ablate CAR T cells. Furthermore, steroids are recommended in patients with adverse neurological effects.

### 5.2. IL-6 Activity Inhibitors

Tocilizumab is a humanized monoclonal antibody against both the soluble IL-6 and membrane-bound IL-6 receptors, inhibiting both classical and trans-IL-6 signaling. After multiple trials demonstrated its efficacy [76,87], it was approved by the FDA in 2017 as the first approach for the treatment of CRS-related toxicities following CAR T-cell infusion. Tocilizumab controls CRS without significant loss of CAR T-cell activity. Improvements occur within a few hours after drug infusion, reducing adjuvant therapy. Advantageous effects of a single injection in patients with CRS induced by CAR T-cell therapy strongly suggest that IL-6 blockade may constitute a new therapeutic approach for an acute, severe, systemic inflammatory response such as CRS. Its most used dosage for CRS is 12 mg/kg for patients weighing <30 kg and 8 mg/kg for patients weighing ≥30 kg. Fever and hypotension are ameliorated within a few hours in responsive patients, whereas it is necessary to continue supportive treatment for several days in some patients. Responsive patients recover from CRS without significant loss of CAR T-cell function. However, a significant number of patients present with resistance to tocilizumab [83].

The effect of tocilizumab against severe CRS-associated neurotoxicity is extremely low, probably due to its limited ability to penetrate the BBB [88]. Some researchers have seen that prophylactic use of tocilizumab does not prevent the development of neurotoxicity [89], whereas others wonder if it is possible to use this drug prophylactically in order to avoid or reduce the symptoms of CRS [22].

Another monoclonal antibody that blocks IL-6 signaling is siltuximab, which prevents the activation of immune effector cells through either the trans or classical mechanisms [50]. Siltuximab has a higher affinity for IL-6 than tocilizumab has for IL6R, making it an attractive tool in managing CRS. The use of siltuximab is encouraged in patients that do not respond to tocilizumab and corticosteroids.

### 5.3. IL-1, IFN-γ, TNF-α, and IL-2 Inhibitors

IL-6/IL6R blockade alone cannot alleviate CRS symptoms completely and would be insufficient in treating severe CRS in patients undergoing CAR T-cell therapy. For patients who become critically ill and do not respond to IL-6, directed therapy targeting IL-1, IFN-γ, TNF-α, or sIL-2 could ameliorate their symptoms. This has led to the use of other cytokine inhibitors, including TNF-α and IL1R inhibitors [29,56,70,80]. Etanercept and infliximab both target TNF-α, which is known to be elevated in CRS, and have been used to treat severe CRS, with varying results [29,80]. Anakinra, a recombinant and slightly altered form of the IL1R antagonist, may be helpful in a subset of patients with increased IL-1α, but this cytokine is also not consistently elevated in patients with severe CRS [70]. Although IFN-γ is consistently elevated very early in severe CRS, it is not thought to be an ideal target due to its role in T-cell proliferation [90].

Finally, other research groups have thought of inserting a “suicidal” gene into the genetic construct used to arm T lymphocytes, a “command” that, when activated from the outside, drives the lymphocytes to commit suicide, self-limiting activity when the CRS becomes life-threatening [76].

## 6. Conclusions

The introduction of CAR T-cell therapy into clinical practice is revolutionizing the treatment of numerous hematological malignancies. This treatment is able to induce prolonged remission in patients in a very advanced stage of the disease and for whom the most common therapeutic options have been exhausted. However, it is essential that life-threatening toxicities, such as CRS, can be managed in an optimal and effective way. Therefore, the goal is to prevent or effectively treat CRS without diminishing the antitumor efficacy. The BiTE construct blinatumomab, a prophylactic procedure including cytoreduction, premedication with corticosteroids, and dose adjustment, appears to be able to reduce the incidence of severe CRS [82]. Considering the type of CAR T cells currently available, the most used sequence of agents to control severe CRS include tocilizumab and high-dose corticosteroids. When these treatments fail, other cytokine inhibitors, such as TNF-α or IL1R inhibitors, are often used. Isolated and severe neurotoxicity is usually, at least initially, treated with corticosteroids rather than tocilizumab.

In this review, we have summarized the pathophysiology, symptoms, and management of CRS associated with T-cell-based therapies utilized in the treatment of hematological malignancies. Although several grading scales and very effective treatment algorithms have been proposed, we would like to emphasize that optimal management of patients is based on the close collaboration of a multidisciplinary team that includes hematologists, neurologists, radiologists, and intensive care specialists, so that mechanical ventilation can be offered if necessary. Until new knowledge on the pathophysiology of CRS allows the use of new and more effective treatments, we think that early intervention by a multidisciplinary team and the use of tocilizumab and corticosteroids remains the best management strategy.

## 7. Future Prospect

In the near future, every effort will be made to balance treatment toxicity and efficacy and to prevent or reduce the symptoms of CRS after infusion of T-cell-based therapies in hematological patients. It will be necessary to act both on the aspects of patient care and in the field of research on that intricate network of different cell types, which constitutes the immune system. These goals can be achieved:(a)Developing even greater cooperation between experts from various fields such as onco-hematology, neuroscience, immunology virology, and ICU: the multidisciplinary approach is an essential requirement for an adequate treatment of patients with CRS.(b)Improving specificity and further unlocking the potential of immunotherapy.

A typical example is the recently developed split, universal, and programmable (SUPRA) CAR system which is composed of an antigen-binding portion, and a universal signal transduction receptor. The system seems able to improve the specificity and controllability [91] of the immune cell engineering strategy. Thus, the SUPRA CAR system has the potential to reduce CRS without reducing the antitumor response [92].

## Figures and Tables

**Figure 1 ijms-22-07652-f001:**
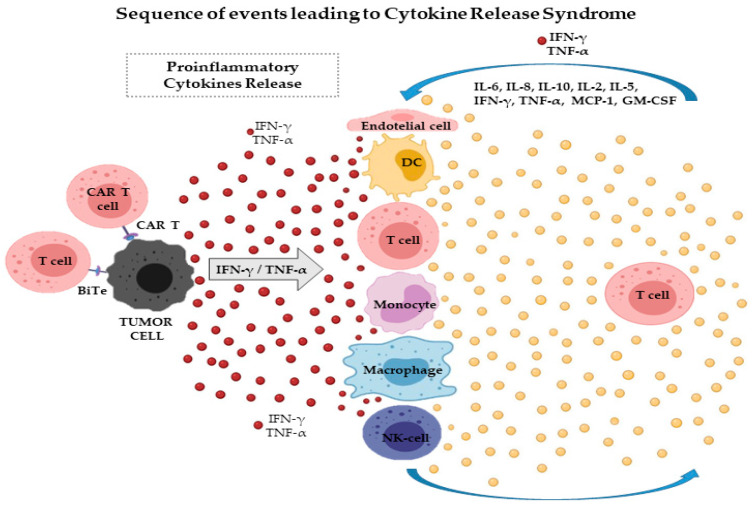
CAR T cells target tumor cells and induce the release of cytokines as IFN-γ or TNF-α, which lead to the activation of bystander immune and non-immune cells as monocytes/macrophages, dendritic cells, NK and T-cell, and endothelial cells. These cells further release proinflammatory cytokines triggering a cascade reaction. Macrophages and endothelial cells produce large amounts of IL-6 which in turn activates T cells and other immune cells leading to a cytokine storm. BiTe: bispecific T-cell engager; CAR: chimeric antigen receptor; IFN-γ: interferon-gamma; TNF-α: tumor necrosis factor-alpha; IL: interleukin; GM-CSF: granulocyte colony-stimulating factor; MCP-1: monocyte chemoattractant protein; NK cell: natural killer cell; DC: dendritic cell.

**Figure 2 ijms-22-07652-f002:**
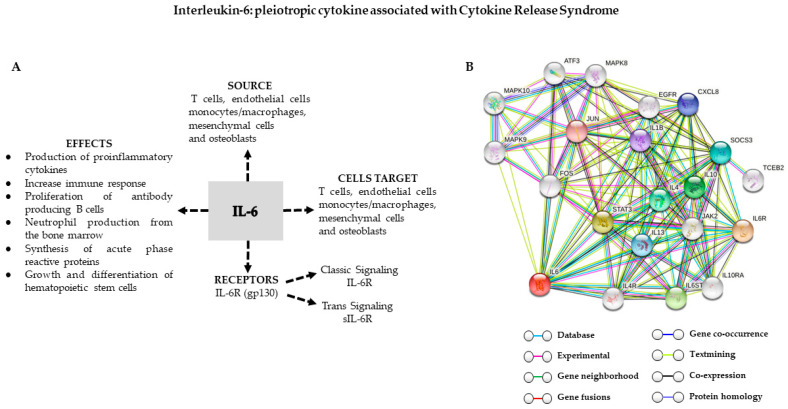
(**A**) Source and biological functions of IL-6. (**B**) Genetic interaction network (String: https://string-db.org) that evaluates pathways and visualizes the connection among target genes according to the literature search. IL6R: interleukin-6 receptor subunit alpha; STAT3: signal transducer and activator of transcription 3; IL6ST: interleukin-6 receptor subunit beta; IL10: interleukin-10; IL4: interleukin-4; SOCS3: suppressor of cytokine signaling 3; IL13: interleukin-13; CXCL8: interleukin-8; IL1B: interleukin-1 beta; JUN: transcription factor AP-1.

**Figure 3 ijms-22-07652-f003:**
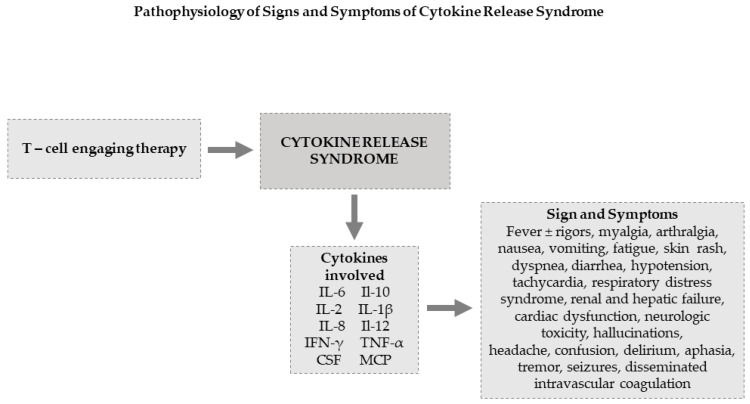
Pathophysiology of signs and symptoms of CRS. IFN-γ: interferon-gamma; TNF-α: tumor necrosis factor-alpha; IL: interleukin; CSF: colony-stimulating factor; MCP: monocyte chemoattractant protein.

**Table 1 ijms-22-07652-t001:** Monoclonal antibodies associated with cytokine release syndrome (CRS) in hematological malignancies.

Antibody	Antigen	Class	Reference
TGN1412	CD28	Human IgG4	[31]
Alemtuzumab	CD52	Human IgG2	[13]
Rituximab	CD20	Murine-human chimeric IgG1	[10,11]
Obinutuzumab	CD20	Human IgG1	[12]
Brentuximab	CD30	Human IgG1	[14]
Dacetuzumab	CD40	Human IgG1	[15]

**Table 2 ijms-22-07652-t002:** Soluble mediators over-expressed in cytokine release syndrome.

Main Cell Source	Cytokines	Type and Function	Reference
Macrophages, epithelial cells	**IL-1**	Proinflammatory alarming cytokine; pyrogenic function, macrophage, and Th17 cell activation	[32,33]
T cells	**IL-2**	Effector T-cell and regulatory T-cell growth factor	[33,34]
Monocyte/macrophages, T cells, endothelial cells, mesenchymal cells, osteoblasts	**IL-6**	Proinflammatory cytokine; pyrogenic function, increased antibody production, growth and differentiation of hematopoietic stem cells, induction of acute-phase reactants	[33,35]
Regulatory T cells, T cells	**IL-10**	Anti-inflammatory cytokine, inhibition of Th1 cells, and cytokine release	[33,36]
	**Chemokines**		
Macrophages, epithelial cells	**IL-8 (CXCL8)**	Recruitment of neutrophils	[33,37]
Monocyte, endothelial cells, keratinocytes	**IP-10 (CXCL10)**	Interferon-inducible chemokine: recruitment of Th1 cells, NK cells, plasmacytoid dendritic cells	[38]
Macrophages, dendritic cells, cardiac myocytes	**MCP-1 (CCL2)**	Recruitment of Th2 cells, monocyte, dendritic cells, basophils	[39]
Monocyte, neutrophils, dendritic cells, NK cells, mast cells	**MIP-1α (CCL3)**	Recruitment of macrophages, Th1 cells, NK cells, eosinophils, dendritic cells, pyrogenic function	[40,41]
Macrophages, neutrophils, endothelium	**MIP-1β (CCL4)**	Recruitment of macrophages, Th1 cells, NK cells, dendritic cells	[40,41]
	**Growth Factors**		
Th1 cells, CTLs, group 1 innate lymphoid cells, and NK cells	**IFN-γ**	Proinflammatory cytokine, activation of macrophages	[42]
Macrophages, T cells, NK cells, mast cells	**TNF-α**	Increasing vascular permeability, pyrogenic function	[43]
Th17 cells	**GM-CSF**	Proinflammatory cytokine	[44,45]
Endothelium and macrophages	**CSF**	Growth and differentiation of neutrophils	[44]
	**Plasma Protein**		
Hepatocytes	**CRP**	Monomeric CRP increases IL-8 and MCP-1 secretion, IL-6 increases CRP expression	[46]
Ubiquitous	**Ferritin**	Primary site of iron storage in cells	[47]

IL: interleukin; IP-10: interferon—inducible protein 10; MCP-1: monocyte chemoattractant protein; MIP: macrophage inflammatory protein 1α; IFN-γ: interferon-gamma; TNF-α: tumor necrosis factor-alpha; GM-CSF: granulocyte colony-stimulating factor; CSF: colony-stimulating factor; CRP: C-reactive protein; Th cell: T helper cell; CTLs: cytotoxic T lymphocytes; NK cell: natural killer cell.

**Table 3 ijms-22-07652-t003:** Cytokine release syndrome (CRS) grading system (Lee et al. [29] partially modify).

Toxicity	Grade
Symptoms are not life threatening and require symptomatic treatment only, eg, fever, nausea, fatigue, headache, myalgia, malaise	**Grade 1**
Symptoms require and respond to moderate intervention. Oxygen requirement < 40 % or hypotension responsive to fluids or low dose of one vasopressor or grade 2 organ toxicity	**Grade 2**
Symptoms require and respond to aggressive intervention. Oxygen requirement ≥ 40 % or hypotension requiring high dose or multiple vasopressor or grade 3 organ toxicity or grade 4 transaminitis	**Grade 3**
Life-threatening symptoms. Requirement for ventilator support or grade 4 organ toxicity (excluding transaminitis)	**Grade 4**

Grades 2–4 refer to CTCAE v4.0 grading [21].

## Data Availability

Not applicable.

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
