# Peer review of "Cytokine Release Syndrome Associated with T-Cell-Based Therapies for Hematological Malignancies: Pathophysiology, Clinical Presentation, and Treatment"

_ijms, 2021, doi:10.3390/ijms22147652_

Round 1

Reviewer 1 Report

In this article, Cosenza and colleagues comprehensively elaborated cytokine release syndrome (CRS) in the pathogenesis and its use for therapeutic utility for the treatment of hematological malignancies.  It is well compiled, nicely presented, and well written. The table and illustrations are quite informative.

The following comments/suggestions to improve the article

  • No information about 1L-17 that plays a critical role in CTCL.
  • Table 2 does not contain any reference as this is derived information from published work
  • Future prospects should be written after the conclusion

Author Response

1) No information about IL-17 that plays a critical role in CTCL.

Yes, you're right. IL-17 plays an important role in the pathogenesis and progression of cutaneous T cell lymphoma. Furthermore, we have done a careful search with the following keywords: IL-17, CRS and pathogenesis, but we have not been able to find articles where CRS is related to IL-17. This interleukin plays a prominent role in autoimmune diseases. However, it does not play a relevant role in CRS induced by T-Cell engaging therapies. For this reason, we have not changed the text of the manuscript

2) Table 2 does not contain any reference as this is derived information from published work

Following the reviewer's suggestions, in Table 2,  we added the references for each soluble mediator (cytokine, chemokine, growth factor) over expressed [ref 32 – 47]. Consequently, the Section “References” at page 12 have been updated. 

3) Future prospects should be written after the conclusion

Following the reviewer's suggestions, after the Section 6: Conclusion at page 11, we have added the Section 7: “Future Prospects” and two references [91, 92]. Consequently, the Section “References” at page 12 have still been updated.

In "Future Prospect" we have added the sentence: "In the near future, every effort will be made to balance treatment toxicity and efficacy and to prevent or reduce the symptoms of CRS after infusion of T-cell-based therapies in hematological patients. It will be necessary to act both on the aspects of patient care and in the field of research on that intricate network of different cell types, which constitutes the immune system. These goals can be achieved:

a) Developing even greater cooperation between experts from various fields such as onco-hematology, neuroscience, immunology virology, and ICU: the multidisciplinary approach is an essential requirement for an adequate treatment of the patient with CRS.

b) Improving specificity and further unlocking the potential of immunotherapy.

A typical example is the recently developed split, universal, and programmable (SUPRA) CAR system which is composed of an antigen binding portion, and a universal signal transduction receptor. The system seems able to improve specificity and controllability [91] of the immune cell engineering strategy. Thus, the SUPRA CAR system has the potential to reduce CRS without reducing the antitumor response [92]”.

Reviewer 2 Report

Dear Authors,

Thank you for submitting this very interesting paper. CRS is indeed a serious problem not only in infectious diseases but also during anti-cancer tratment. I have only some minor comments about the text:

In title, you state your main iterest will be CRS connected to hematological malignancies therapy. However, in the main text there is only a little mention about it, in Section 2: Pathophysiology. Please elaborate more on connection between hematological malignancies treatment and occurence of CRS.

Table 2 - please rephrase "More Expressed" as either "Over-expressed" or similar in table title

Figure 3 - there is no BiTe in figure, so no need to explain the abbreviation. Similarly, the other abbreviations were already explained previously in Figure 2 description and main text, no need to do it again.

Author Response

1) In title, you state your main interest will be CRS connected to hematological malignancies therapy. However, in the main text there is only a little mention about it, in Section 2: Pathophysiology. Please elaborate more on connection between hematological malignancies treatment and occurrence of CRS.

Following the reviewer's suggestions, in Section 2: Pathophysiology at pag 2 - line n 86, we added the sentence:

“As clearly illustrated in Figure 1, T-cell engaging therapies target tumor cells and induce release of cytokines as IFN-γ or TNF-α, that lead to the activation of bystander immune and non-immune cells as monocytes/macrophages, dendritic cells, NK and T-cell, and endothelial cells. These cells further release proinflammatory cytokines triggering a cascade reaction. Macrophages and endothelial cells produce large amounts of IL-6 which in turn activates T cells and other immune cells leading to a cytokine storm”

2) Table 2 - please rephrase "More Expressed" as either "Over-expressed" or similar in table title

Following the reviewer's suggestions, in title of Table 2, pag 3, line 105, we replaced “more expressed” with “over expressed”

3) Figure 3 - there is no BiTe in figure, so no need to explain the abbreviation. Similarly, the other abbreviations were already explained previously in Figure 2 description and main text, no need to do it again.

We have removed the abbreviation of BiTe from the legend of Figure 3
